

# Research on intelligent file management system: a design strategy based on RFID technology and improved AICT algorithm

Shan Ge

Department of Archives, Wuhan University of Technique, Wuhan, China

## ABSTRACT

In this modern era of technology and digitalization, keeping track of a manual file system in the office environment is a challenging task. This research proposes a radio frequency identification (RFID) technology to improve conventional file management systems. The proposed system includes advances in information and communication technology (AICT) algorithm that addresses tag collision problems, resulting in increased data collection and reduced communication traffic. The system consists of modules such as information collection, file management, and user management. Each is analyzed from various aspects. Simulation results show that the AICT algorithm outperforms the improved collision tree algorithm by increasing recognition efficiency by 10% and reducing communication traffic by 30%. Moreover, the proposed approach provides a simple and convenient way to manage files in real-time meeting the needs of modern times. The AICT algorithm excelled the other algorithms regarding recognition efficiency.

## INTRODUCTION

The economy's rapid development has led to explosive growth in knowledge, and archives have become an important carrier for storing knowledge (*Cai et al., 2021*). Conventional file management has drawbacks, such as being time-consuming, labor-intensive, and prone to file damage and loss due to requiring a lot of storage space (*Kim, 2019*). Barcode technology addressed some of these issues but had limitations such as storing less information, being easily damaged, and having poor anti-pollution capabilities.

Electronic file management systems have improved work efficiency and ensured the safety of files by backing up original paper files. However, large enterprises and companies with many files cannot convert all files to electronic storage in a short period (*Tian et al., 2021*). As a result, physical file management remains the only option, making it necessary to develop file management systems that meet the needs of the current time (*Zhang & Zhang, 2020*).

Corresponding author
Shan Ge, gesh@whut.edu.cn

Radiofrequency identification (RFID) technology, one of the key technologies of the Internet of Things (IoT), emerged at the end of the last century and has rapidly developed in recent years (*Bibi et al., 2017*). It uses radio frequency antennas to communicate, exchange data, identify targets, and integrate signal transmission, automatic identification, radar, and other technologies concurrently. RFID technology has strong storage capacity, higher safety, outstanding anti-pollution ability, and has been maturely implemented in various fields (*He, Linna & Fangfang, 2022*; *Ming et al., 2022*; *Labonnah et al., 2021*). Its application can resolve many problems caused by conventional file management methods and achieve automatic and intelligent management of files.

The application of RFID technology can reduce the workload of managers, reduce the risk of file loss and leakage, speed up the collection efficiency of archives, improve retrieval efficiency, and simplify the management process. In addition, RFID technology is one of the basic technologies needed to realize future smart archives, emphasizing the necessity of research on RFID-based file management systems.

Libraries have similar social functions to archives, and their management methods and functions provide valuable insights. Some countries have developed a relatively complete system for researching RFID libraries. For instance, the Bukit Batok Public Library in Singapore used RFID technology as early as 1998 to realize book inquiry, borrowing and returning, and automatic book sorting (*Pan, Pan & Devadoss, 2008*). The Farmington Community Library and Rockefeller University Library also began using RFID technology in 1999 to assist library management. By 2002, hundreds of libraries in the United States had adopted RFID technology (*Guo et al., 2023*). The National Library of Singapore's cooperation with the post office is an excellent example of integrating the RFID field and conventional field applications.

The motivation of the research is to propose an optimized version of the collision tree algorithm, called the AICT algorithm, based on continuity to manage files more effectively so that tag collision problems are addressed better, resulting in an increased data collection process and reduced communication traffic. To present the contribution of the proposed algorithm, in-depth theoretical analysis and simulation are conducted to demonstrate that the proposed algorithm outperforms the conventional RFID anti-collision algorithm in terms of time spent and space complexity. The effectiveness of the proposed algorithm remains significant even when the location and number of tags in libraries and archives change significantly. Additionally, the distribution of tag identifiers has little effect on the algorithm's performance. The data acquisition rate of tags and accuracy are significantly enhanced. Thus, it is a suitable choice for the data acquisition module of the archive management system. More up-to-date research can be found (*Sastrodiharjo & Khasanah, 2023*; *Takale et al., 2019*; *Li, 2022*).

The outline of the manuscript is constructed as follows: 'The Improvement of the RFID Tag Anti-Collision Algorithm' presents the proposed algorithm. 'Realization of the RFID File Management System' presents the simulation of the proposed algorithm based on three different scenarios. Different realizations of the proposed file management system based on information collection and processing, file management, personal management, and database modules are discussed. The research is concluded in the 'Conclusion' section.

# THE IMPROVEMENT OF THE RFID TAG ANTI-COLLISION ALGORITHM

The RFID tag anti-collision algorithm is proposed. While the basic binary tree algorithm can identify all tags, it has low identification efficiency and requires a large transmission volume in the communication process. As the unique identification code (UID) length of the electronic tag increases, conventional algorithms cannot meet current RFID identification requirements. Therefore, a new algorithm is needed. This article proposes a continuity-based AICT algorithm to improve the available collision tree algorithm.

## The description of the improved algorithm

The proposed AICT algorithm extends the communication technology (CT) and information and communication technology (ICT) algorithms, respectively. While the ICT algorithm improves identification efficiency by reducing the number of queries through binary certainty, it has some limitations and is only effective when tag UIDs are continuous. Additionally, the communication delay and data transmission volume remain high. To address these issues, two improvements were made.

Firstly, the identification of the collision is realized by implementing a method called Manchester encoding (ME). Data is digitally encoded in a binary form characterized by either 0 or 1. Thus, data transition is characterized from one voltage state to another one. Several encoding approaches differ from ME since the bit state of the data is characterized by using the voltage states to reduce the transmission of non-collision bits, which account for most bits in the RFID identification system. The collision bit is then locked through preprocessing, forming the UID for which the tag participates in the identification process.

This reduction of traffic in the communication process also serves to improve recognition efficiency. Secondly, the reader utilizes an adaptive query method to handle different distributions of tag UIDs.

The AICT algorithm utilizes a binary tree search when $-9 \leq C_d \leq -7$, a quadtree search when $-7 < C_d \leq -5$, and an octree search when $-5 < C_d$. By combining binary tree search with multi-ary search, the AICT algorithm can effectively reduce collision time slots in the communication process and improve the recognition efficiency of the system. The concept of continuity defines the probability of continuous UIDs during each recognition, providing a framework for the AICT algorithm (*Yeo & Hwang, 2010*).

## The instruction and execution processes of the improved algorithm

The AICT algorithm's communication process is divided into five states: (1) Success state, where the reader and tag successfully communicate and identify each other; (2) 1-bit collision state, where only one bit collides during identification; (3) continuous state, which uses the ICT algorithm when $-9 \leq C_d \leq -7$; (4) discontinuous state, which is split into decision support system (DSS) state when $-7 < C_d \leq -5$ and double sideband (DSB) state when $-5 < C_d$; (5) gap state, where there is no response from any label and the packet label does not exist. Table 1 provides instructions for each state.

The AICT algorithm consists of consecutive steps as follows:

**Table 1 Algorithm execution command information.**

| Command | Information |
|---|---|
| REQUEST(s(m)) | For interdisk instruction |
| | m is the query length, where $0 \leq m \leq k$ |
| SELECT(PID) | Select the instruction |
| | PID is the new UID composed of the tag lock collision bits |
| READ-DATA | Read instruction |
| UNSELECT (UID) | Sleep instruction |

Step 1: The reader broadcasts instructions (11...1) to all tags in the valid identification field. After reading the data, the process ends if only one tag is present and successfully identified. If a collision occurs, proceed to the next step.

Step 2: The reader preprocesses the collided tag and resends the command. The tag locks the corresponding collision bit based on the command to form a PID identifier. Step 3: The reader selects the appropriate query method based on the continuous range of the tag and stores the query prefix parameter in the stack. There are four types of tag statuses and query methods: (1) Success status or 1-bit collision status: The reader recognizes the tag and sends the READ-DATA command. (2) The tag responds with data and sleeps after receiving the UNSELECT (PID) command. (4) Gap state: The reader reads a new query prefix from the stack and continues execution. (4) Continuous state (Cd>-3): The reader recognizes this according to the execution process of the ICT algorithm. (5) Discontinuous state: In the DSS state, the reader generates four new query prefixes (00, 01, 10, 11), pushes them into the stack pool, and performs a quadtree search. In the DSB state, the reader generates eight new query prefixes (000, 001, 010, 011, 100, 101, 110, 111) and pushes them into the prefix pool. The reader then performs an octree search.

Step 4: After reading the query parameter s(m) from the stack pool, the reader sends the REQUEST(s(m)) command to identify the tag.

Step 5: The tag compares the preprocessed PID with the query prefix s(m). If the first m bits match, it sends the remaining bits (k-m) to respond. If there is no response, discard the query parameters and return to step 3.

Step 6: After reading a round of tags, loop through the above steps until all label recognition is complete.

## Theoretical analysis of the improved algorithm

The improved AICT algorithm's performance is theoretically analyzed by employing metrics of request count, recognition efficiency, and traffic. Here, 'd' represents the query depth of the AICT algorithm.

(1) Number of queries and recognition efficiency:

Assuming that the AICT algorithm's query mode is L (where $L = 2, 4, 8$), and the number of tags to be recognized is N. Considering the search depth to be k, the identification probability of an electronic tag is calculated by employing the Eq. (1):

$$P(k) = (1 - 1/L)^{k-1}. \tag{1}$$

The query depth k is defined by Eq. (2).

$$E(d) = \sum_{k=1}^{\infty} [1 - P(1)] = 1/(1 - 1/L)^{N-1}. \tag{2}$$

The average number of slots, T, required is delineated by Eq. (3).

$$T_L = E(d)L = L/(1 - 1/L)^{N-1}. \tag{3}$$

Assuming the label after preprocessing has eight digits, the AICT algorithm determines the query method based on the degree of continuity. Specifically, when $-5 < C_d$, T8<T4, and when $-7 < C_d \leq -5$, T4<T2, where T2, T4, and T8 denote slots and $C_d$ represents. When compared to the ICT algorithm, the AICT algorithm requires fewer time slots and tag queries due to its automatic determination of the query method based on different degrees of continuity. This reflects the superiority of the AICT algorithm over the ICT algorithm. On average, the collision tree root node and non-leaf nodes have seven labels to identify. The total number of required queries can be calculated in Eq. (4):

$$T_{AICT} = T_2 + T_4 + T_8 = 41N/21 - 2M + 1 + \sum_{i=0}^{(N-7d)/3} 4^i + \sum_{i=0}^{[\log 8N/7]} 8^i. \tag{4}$$

Eq. (5) can be obtained from the definition of recognition efficiency e.

$$e = N/T_{AICT} = N \cdot (41N/21 - 2M + 1 + \sum_{i=0}^{(N-7d)/3} 4^i + \sum_{i=0}^{[\log 8N/7]} 8^i)^{-1}. \tag{5}$$

(2) Traffic:

In the query process of cycle i, let $L_{com}$ be the command word sent by the reader, $L_{se\_i}$ be the query parameter sent by the reader, and $L_{rep\_i}$ be the data sent when the tag responds, then the traffic C(N) of the AICT algorithm is defined by Eq. (6).

$$C(N) = \sum_{i=1}^{T(N)} (L_{com} + L_{se\_i}) + \sum_{i=1}^{T(N)} (L_{rep\_i}.) \tag{6}$$

T(N) represents the number of cycles required to identify N tags. Then, in a recognition cycle, the preprocessing tag length is composed of $L_{se\_i}$ and $L_{rep\_i}$, and the length of the preprocessing tag UID is $L_{pre\_i}$, which can be obtained by

$$L_{pre\_i} = L_{se\_i} + L_{rep\_i}. \tag{7}$$

By the utilization of Eq. (7), the communication volume of the AICT algorithm is defined by

$$C(N) = (41N/21 - 2M + 1 + \sum_{i=0}^{(N-7d)/3} 4^i + \sum_{i=0}^{[\log 8N/7]} 8^i) \cdot (L_{com} + L_{se\_i} + L_{rep\_i}). \tag{8}$$

### The simulation of the RFID tag anti-collision algorithm

Simulation experiments were conducted to compare the performance of five anti-collision algorithms, namely, advances in information and communication technology (AICT), quick time (QT), database system (DBS), communication technology (CT), and information and communication technology (ICT). The experiments considered three different tag UID distribution scenarios based on continuity. These scenarios were defined as continuous, even, and partially continuous with a 91dC degree of continuity. The algorithms were evaluated based on average query times, recognition efficiency, and communication volume. The experimental parameters included one reader, 96-bit tag length, 96kbps data transmission rate, and a tag count ranging from 4 to 2048. Monte Carlo sampling was used for tag selection, and MATLAB 2017 was used to create simulation graphics. Results were based on an average of 50 independent experiments.

### The analysis of Scenario 1

In Scenario 1, several RFID algorithms were analyzed based on their performance curves for the number of requests, recognition efficiency, and traffic. Among the algorithms, the AICT algorithm stood out as it recognized a label 0.8 times on average, which is higher than that of the ICT and CT algorithms which require more time, respectively. The QT algorithm, on the other hand, had an inverse relationship between the number of queries and the number of tags, respectively.

Regarding recognition efficiency, the AICT algorithm had a higher increased recognition efficiency of 110%, which is 10% and 70% higher than the ICT and CT algorithms, respectively. However, the maximum recognition efficiency of the QT and DBS algorithms was only 50%. Although the AICT algorithm has obvious advantages due to preprocessing that shortens the length of the label, communication volume was reduced by only 10% when compared to the ICT algorithm due to a limited reduction in the number of queries.

### The analysis of Scenario 2

In Scenario 2, the ICT and CT algorithms required the same number of queries to identify a label, and recognition took two times. The AICT algorithm stood out again with clear advantages, as it required only 1.4 times the number of queries and had a recognition efficiency 20% higher than that of the ICT and CT algorithms. The DBS and QT algorithms had similar recognition times as those in Scenario 1.

Like in Scenario 1, the AICT algorithm reduced the number of transmission bits after preprocessing the tag length. However, the significant reduction in the number of queries meant that the average communication volume of the AICT algorithm was 30% lower than that of the ICT and CT algorithms and far lower than that of other algorithms.

### The analysis of Scenario 3

As the degree of continuity increased in Scenario 3, the number of requests for the AICT algorithm decreased while recognition efficiency increased with an increase in the number of labels. The minimum number of queries was 0.7, and the recognition efficiency increased to 95% from 70% respectively.

When the number of tags is large enough, for example, in 2048, the communication traffic was significantly reduced because more tags were continuous as their UID distribution became uniform or partially continuous. Thus, the AICT algorithm query times and traffic were significantly reduced, improving the identification efficiency of the RFID system while saving transmission bandwidth.

The section conducted three scenarios of labeled UID distribution and compared four different algorithms to analyze their efficiency. The algorithms used were AICT, DBS, QT, CT, and ICT, respectively. The comparison was based on the average number of requests, recognition efficiency, and communication volume. MATLAB 2017 simulation tool was used. The results showed that the AICT algorithm outperformed the other algorithms regarding recognition efficiency when the tag UIDs continuously improved it by 10%. However, when the tag UIDs were non-continuous, the improvement was over 20%. Regarding reducing traffic, the AICT algorithm used a preprocessing method for label numbers, which reduced the length of labels and decreased traffic by 30%. Overall, the study suggests that the AICT algorithm is an effective solution for improving the efficiency of label recognition and reducing traffic in scenarios where tag UIDs are either continuous or non-continuous. A laptop STRIX Z270H Gaming with a CPU Intel® Core™ i7-7700K CPU 4.20 GHz (8 Cores), memory 15Gb, and GPU Nvidia GeForce GTX 1080 is used.

# REALIZATION OF THE RFID FILE MANAGEMENT SYSTEM

## Information collection and processing module

The RFID file management system relies heavily on the information collection and processing module, which is a fundamental component and plays an essential role in ensuring the functionality of other modules. The article focuses on the archives management system's information collection and processing module, which comprises five key parts: archives storage and retrieval, archives tracking, archives borrowing and returning, personnel management, access control management, and related data collection. Each of these components will be elaborated upon in detail below.

(1) Information collection from archives in and out of the warehouse:

Collecting information on archives entering and leaving the warehouse involves two main aspects. Firstly, when new files are stored in the filing cabinet, management personnel must affix labels to them. The system then automatically generates information such as the file number, warehousing time, and storage location, which is recorded in a data sheet. Additionally, managers enter information such as the company department, confidentiality level, and storage period into the corresponding database table. Secondly, archives are typically transferred out of the library according to a file transfer letter provided by the user.

Managers use search tools to locate the corresponding file and record information such as the file transfer person, recipient, and release time into the database before releasing the file. To streamline this process, an electronic tag system can be used. By placing the file on a designated table with an embedded reader antenna, the label information is automatically read without needing a handheld device. This method is convenient, saves space, and eliminates interference from other items.

(2) Information collection of archives in place:

Collecting information about the presence of files involves real-time detection of whether a file is in a specified location. This is done to ensure the security of file information and reduce the hassle caused by misplaced files, making it easier for managers to locate or count file information while reducing the possibility of file losses.

To improve efficiency and save costs, the article adopts passive UHF electronic tags with small size, low cost, repeated use, and fast reading and writing speeds, which are pasted on the file bags. In-place information collection for archives includes file bag and file box positioning.

Since numerous file bags are in the file box, ordinary high-power antennas cannot effectively read the electronic tags on densely packed file bags. Therefore, setting reasonable positions for labels on file boxes and file bags is vital. The manuscript designs file bags that are 242 mm × 26 mm × 335 mm in size and uses UHF tags that are 45 mm × 45 mm in size. Care should be taken when pasting to ensure that the direction and position of the electronic tags are consistent. The distance from the portfolio's bottom edge and inner edge should be 110 mm and 70 mm, respectively.

In this study, portfolio information is collected through a four-channel multi-channel reader and an antenna group installed on each floor of the filing cabinet. The four outputs of the four-channel multi-channel reader are connected to the four-layer antenna group of the filing cabinet. By doing so, electromagnetic interference generated in identifying tags by too many readers can be avoided. The schematic diagram of reader and antenna installation in this file system is presented in Fig. 1.

The multi-channel reader/writer follows a detection command from the file management system. It reads each layer of file bags in a cycle and switches to the next layer after reading one. Once all layers are read, the cached data is returned to the background management system for analysis and processing. If the system finds any misplaced or missing files, it prompts the management personnel to take timely action. The information collection and processing method for file boxes is similar to that of file bags. However, while pasting, it is important to maintain the direction and position of the electronic label. A one-way reader and antenna module are used for file boxes to differentiate between file bags and file boxes during reading. The antenna module's size matches the file box's thickness, ensuring accurate readings.

(3) Information collection of file loan and return information:

The process of collecting information regarding loans and the return of files involves two stages. In the first stage, the reader reads the label information of the borrowed file when it is issued to a borrower. In the second stage, the reader confirms the return of the file by reading its electronic tag information. The status of the archives in the system is updated accordingly to reflect the changes in their borrowing and returning status.

This collection of information is similar to the process of tracking files entering or leaving a warehouse. However, in this case, an all-in-one machine that integrates a reader with an RF antenna is used to collect the data. This machine offers a more convenient and faster way of processing information for a small number of files.

(4) Information collection of personnel information:

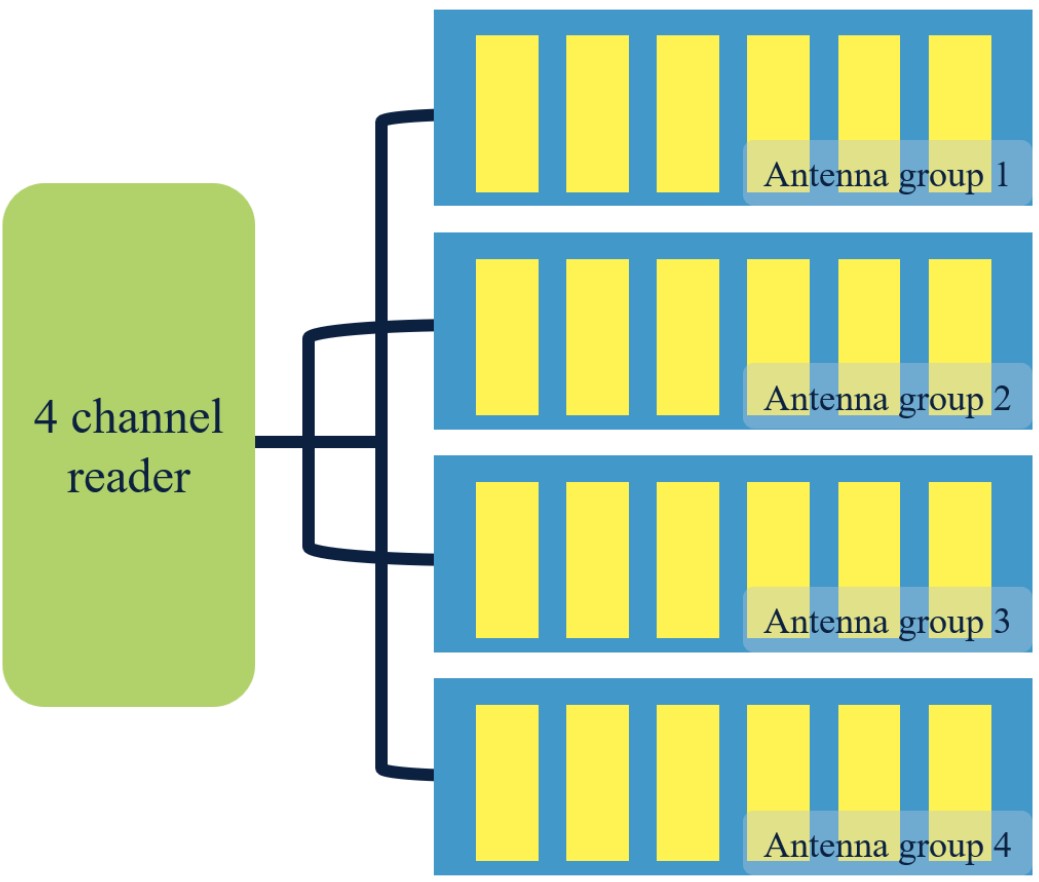

**Figure 1** Diagram of antenna layout of file reader.

The system primarily collects two types of information: the registration and card issuance details of both administrators and borrowers and the identity verification of users during file borrowing and return. During user registration and card issuance, the reader scans the identification information on the card and transmits it to the background management system. The card number is then linked with the registered borrower's information. When the borrower returns a file, the reader scans their ID card information and retrieves the associated file borrowing information from the computer management system. The file return process is completed after the management personnel ensure that the information is correct. The M1 card, which adheres to the ISO14443A international agreement and is compatible with most employee cards, is used by the borrowers in this system. This card type allows for the collection of personnel information.

(5) Information collection of access control management:

The file management system contains an access control module as an additional security layer. This module is linked to the access controller, reader, and alarm device. When a file passes through the access control device, the reader reads the information and then sends it back to the computer management system to verify its legitimacy. If the file is

unauthorized, an alarm command is issued by the computer management system to give access to the controller, which activates the driving alarm device to alert management personnel. This process helps to safeguard the files.

## File management module

The file management module is a crucial component of this system, implementing both B/S and C/S software architectures to cater to the needs of managers and regular users. The client management system is an all-encompassing tool for managing file data, with features such as file query, entry and exit management, borrowing and return management, personnel management, and system maintenance primarily utilized by managers (*Tu & Piramuthu, 2017*). On the other hand, the WEB side management system offers remote accessibility for users. This system allows users to check if the necessary files are available, view their borrowing records, and reserve files. Therefore, the WEB terminal management system fulfills three primary functions: file query, user management, and file reservation. The file query management function is the most frequently used feature among these functions. The following section will introduce the file query function by combining the WEB and client file query interfaces.

(1) File query management:

The web-based file management system is constructed by using Eclipse and Tomcat servers, with the simple system manager (SSM) approach adopted as its software web framework. The SSM method combines three frameworks, Spring, Spring MVC, and MyBatis, which are commonly used in modern Java software architecture.

The Spring MVC is a lightweight web framework that operates on a request-driven basis. It simplifies the development process of web pages considerably (*Zhu et al., 2017*). The execution process consists of several steps. The front controller analyzes the user's requests and determines which page controller will process them based on the requested content, such as the URL. The page controller collects and binds the request parameters into a command object, which is then delegated to the business object for processing. After the business object completes processing, a model view (model and view) is returned. Finally, the front-end controller presents the rendered model view to the user, and thus the entire request–response procedure ends. Figure 2 displays the Spring MVC workflow diagram.

MyBatis is a lightweight Java-based persistence layer framework that offers flexibility and independence from third-party dependencies and simplifies database operations such as adding, deleting, modifying, and checking by requiring only SQL statements in XML files that can be called through software codes. MyBatis also supports object-database relational mapping, making database maintenance less complex. To illustrate the workflow of MyBatis, Fig. 3 depicts it.

The application loads the MyBatis operating environment through the user's configuration file SqlmapConfig.xml, creates a session factory, and the session factory creates a session. If the transaction created by the session is normal, it will be submitted to the database, and operations such as adding, deleting, modifying, and checking the database will be performed. If the transaction is abnormal, it will be rolled back to the session, and the database operation will fail. SSM is a lightweight and powerful WEB development

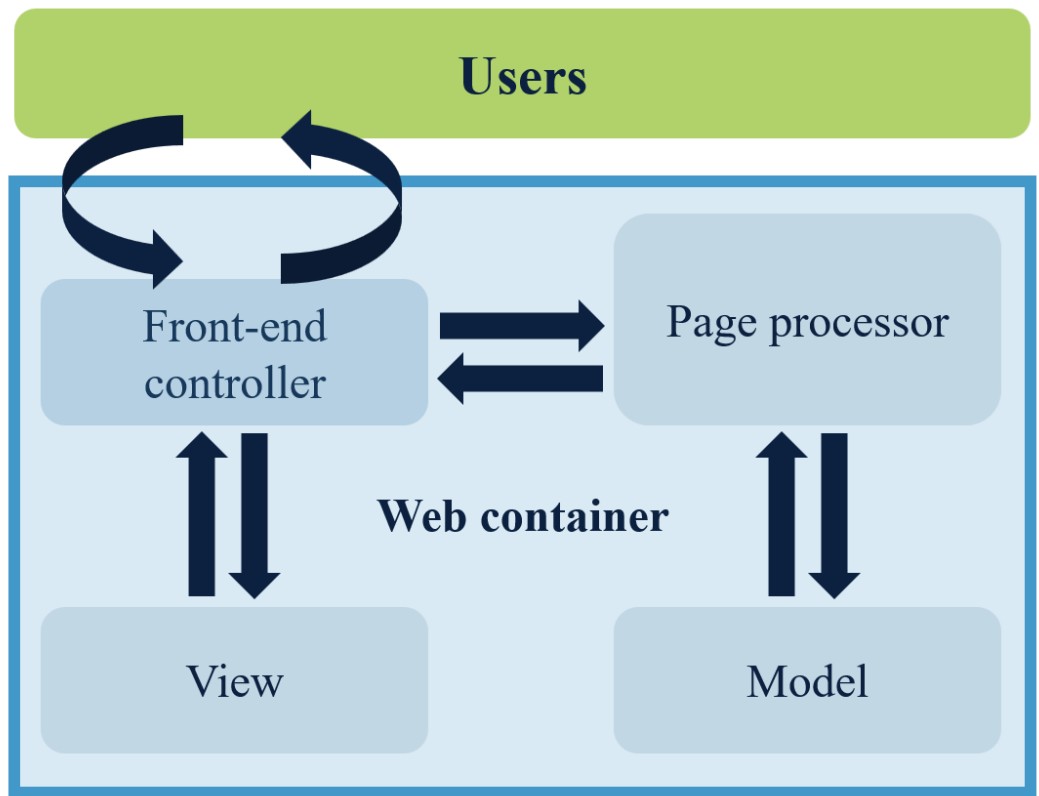

**Figure 2 Flowchart of how Spring MVC works.**

framework that can help developers clarify business logic, reduce development difficulty, and facilitate expansion and maintenance.

The platform allows users to search for files based on their numbers and perform more specific searches by utilizing the file name, type, and storage time filters. If a desired file is found, the user can schedule an appointment to access it by providing relevant personal information, such as name, pickup time, phone number, and email address. The management team then reviews this information. Users can check their appointment status in the mailbox located on the upper right-hand side of the interface, regardless of whether it was successful or not.

(2) File entry and exit management:

The term "files out of the library" typically pertains to individuals or departments applying to borrow files. The approval process for releasing these files is contingent upon the specific situation. Only approved requests will be allowed to remove files from the library. To initiate this process, the administrator will first verify whether the requested file is eligible for release from the warehouse. Once it is confirmed that the file can be released, they will check its current availability status. Following this, they will update both the label

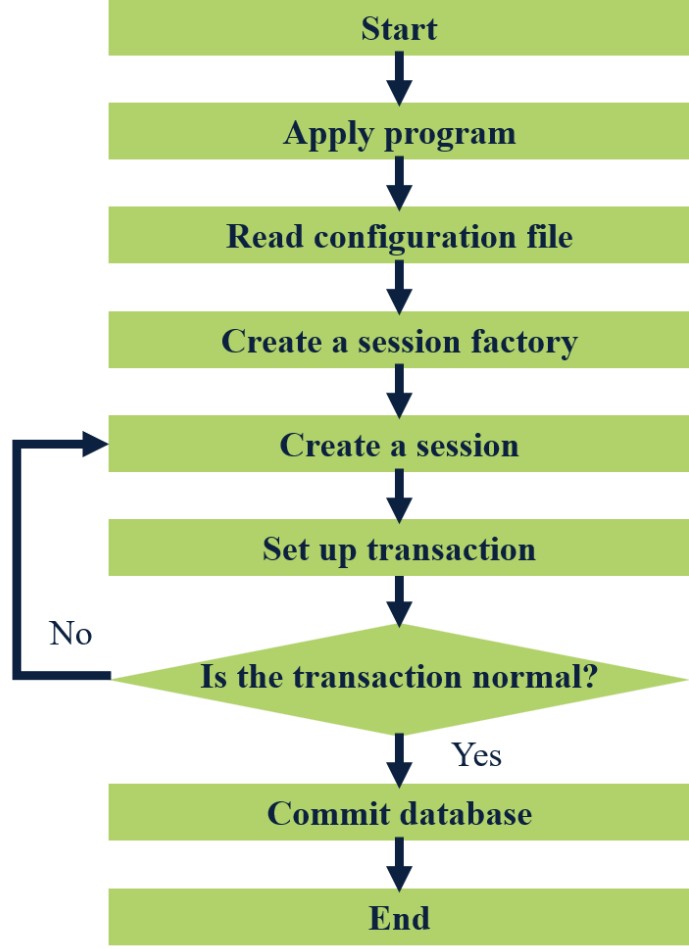

**Figure 3    Flow chart of MyBatis working.**

information and database records associated with the file. Figure 4 illustrates the detailed steps involved in the file-out process from the warehouse.

The storage of files involves adding new files to the file system. Firstly, the manager places the file in a designated bag and labels it appropriately. The label is then scanned by a reader, which automatically generates a unique file number, storage location, and additional information for the new file in the system.

Furthermore, the management staff also creates a backup of the file information in the database. Once completed, the file can be put into the file cabinet, signifying the completion of the storage process. A visual representation of the file storage flow chart is depicted in Fig. 5.

(3) File borrowing and returning management:

The file lending management is a commonly used function in the system, consisting of two sub-modules: file borrowing and file return. In the archives borrowing module, users can apply to borrow files by making an appointment through the WEB terminal,

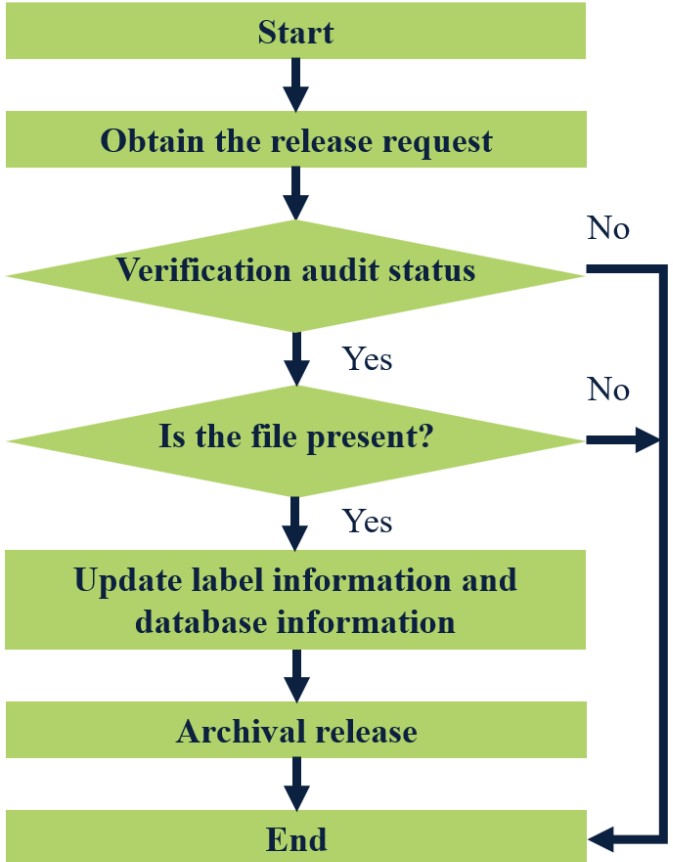

**Figure 4  Flow chart of archival release.**

submitting a request through their department, or as an individual. Once the application is received, the administrator checks the user's credit score and proceeds to locate the file through the archives query function if it is satisfactory. After the file is checked whether it is available for borrowing, the file label and database information are updated to indicate that the file has been borrowed successfully. The file return management module has three primary functions: confirmation of returned files, returning files, and generating return records. When the user returns a file, the manager inspects its condition and confirms its correctness. Then, using a reader placed on the table, the label information of the file is obtained to determine its storage location. Afterward, the system generates a file return record and updates the status of the file bit and database information. Finally, the file is returned to its original storage location to complete the file return operation.

(4) File statistics management:

This module enhances file management by incorporating statistical functions related to files. The implemented specific statistical features may vary depending on the department's needs, as the demand for statistics is not constant. The module includes key functions

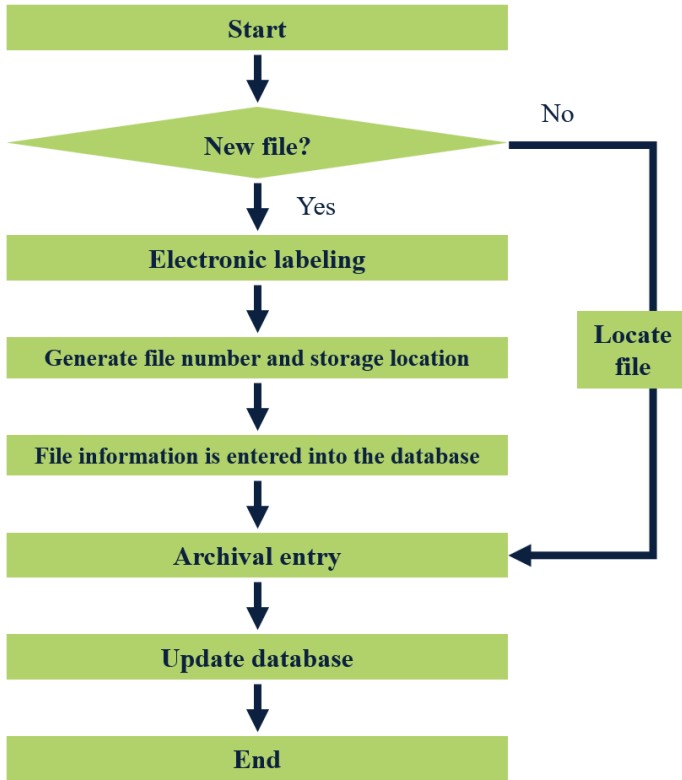

**Figure 5** **Flow chart of file entry.**

such as online file printing, exporting file information, and generating charts that provide valuable insights.

Online printing enables management personnel to quickly print file information, file cabinet details, file entry/exit records, and file loan/return records, amongst others. This module can export archive-related information into electronic documents to cater to varying departmental requirements.

To facilitate analysis of archive management work, this module is equipped to generate icons or visual representations of relevant archive management information. For instance, managers could determine which months have a high frequency of file borrowing through the line chart feature depicting file borrowing records. Similarly, with the help of file information pie charts, they can analyze the classification and proportion of files.

## Personnel management module

The personnel management module provides two key functions: user management and reputation query. User management is categorized into system administrators, common administrators, and common users. Appropriate permissions are assigned to various management personnel to achieve a hierarchical management system for the file system. Reputation points are used to evaluate users' borrowing and repaying behavior based on

certain standards, restricting their actions to ensure the security of the files (*Zhu et al., 2017*).

The system administrator has complete control over the system and holds the highest authority, enabling them to create ordinary user accounts, assign permissions, and modify or delete highly confidential files as needed. They are also responsible for maintaining the system and overseeing the operation rights of all other modules.

The system administrator authorizes ordinary administrators and is in charge of the system's daily operation, handling tasks such as file entry and exit management, borrowing and returning management, statistical management, and database management. Regular administrators play a crucial role in the system's regular usage.

Ordinary users refer to archive borrowers who can only perform basic queries and borrowing operations without system management file modification or deletion privileges. The system administrator provides their authority, and they must utilize their borrowing card or linked employee card to complete corresponding operations. Borrowers may also apply for system access through the WEB file management system. Figure 6 illustrates the user management interface of the client file management system.

The client file management system implements a reputation point mechanism to manage user's behavior. Users receive a certain percentage of reputation points, which can be deducted if they exhibit irregular behavior. If a user fails to return a file on time, 10 points will be subtracted, and if a file is damaged, 30 points will be deducted. If a user's credit score falls below 80, they are disqualified from borrowing files. Administrators must review a user's credit score before allowing them to borrow files. Only users with good reputations may borrow files.

The web-based file management system functions similarly to the client system. However, there are some differences. The web system is only available for ordinary users to register. When a user submits personal information, the system administrator reviews it. After the review is finalized successfully, the administrator assigns permissions and sends an email notification to the user. Users cannot modify or view their credit points.

## Database design

Database design is a crucial aspect of software development as it directly impacts the software system's scalability, data storage, and overall business development. A well-designed database should have fast response times, strong scalability, high stability, and easy handling of complex business data-logic relationships. In addition, both file management WEB end and client systems can share a database, making it even more critical to get it right.

To achieve an effective database design, this research outlines six key steps: 1. Requirements analysis stage: This stage involves adopting a top-down approach to understand the general workings of the system, identify user needs, and consider future expansion possibilities. It is also important to analyze the various requirements of the file management system at this stage. This phase is typically the most time-consuming in the database design process. 2. Conceptual structure design stage: This step requires abstracting a core object for real affairs using an inside-out methodology. Then, all conceptual models

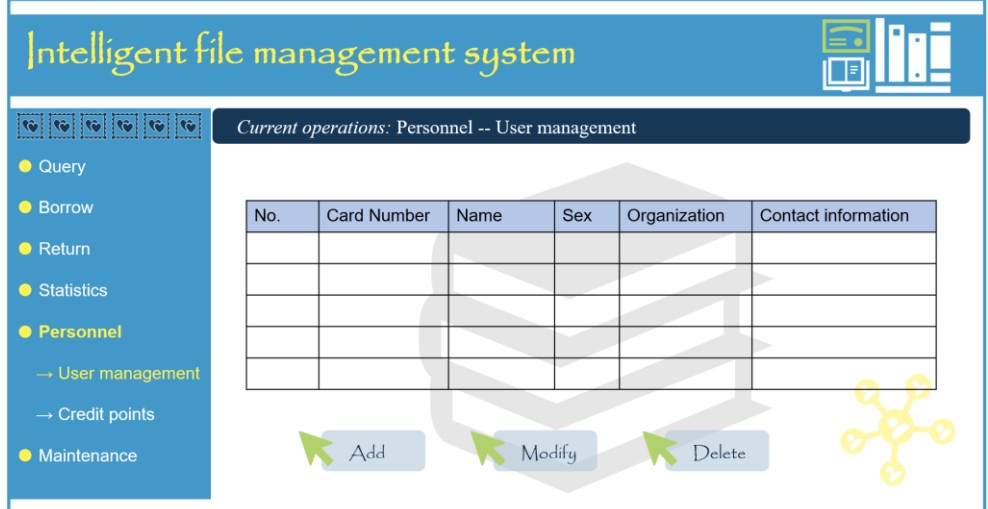

**Figure 6** **User management interface of intelligent file management system user port.**

in the transaction are established based on the logic between transactions, which are then represented using ER diagrams.

3. Logical structure design stage: This stage aims to transform the conceptual design model into a logical structure model supported by a specific database. 4. Physical structure design stage: This stage aims to select a physical structure with high time and space efficiency to store the logical structure model while also analyzing the method and storage method provided by the physical structure used to ensure it is suitable for the system's data requirements. 5. Database implementation stage: The designers establish the database according to the physical design results and import data into the database through the database language, such as SQL. They must ensure that the trial operation conforms to the grammar specification of the database language and compile and debug the application program to test whether the database meets the system requirements. 6. Database maintenance stage: After the database is in normal operation, it is crucial to continuously adjust and modify it to maintain its efficient and stable operation.

An effective database design is critical for efficiently operating software systems. By following these six steps, designers can create a database that meets the system's requirements, is scalable, has fast response times, and is easy to maintain.

The file management system stores all its data in the MYSQL database, where the database for the document management system is named "docment_manage". All tables used by the system are kept within "docment_manage", which includes the following tables:

Administrator information table: "info_administrators"; department information table: "info_department"; file information table: "info_documents"; file cabinet information table: "info_repertory"; User table: "info_users"; file entry and exit record table:

"log_warehousingOutbound"; file borrowing and returning information registration form: "log_returnBorrow"; user reservation registration form: "info_userOrder"

This section focuses on the implementation of an RFID-based file management system. It begins by introducing the archives information collection and processing module, which has five parts: collection of archives entry and exit information; collection of archives in-place information; collection of archives borrowing and return information; collection of borrower information; and collection of access control management information. Next, the section details the functions of the file management module and user management modules combined with the software interface of the file management system client and WEB end. Finally, it explains the database's design principles, steps, and specific table structures.

## CONCLUSION

The present study focuses on designing and implementing a file management system based on RFID technology. This technology has gained immense popularity recently and has been extensively used in various domains.

The key contribution of this article lies in proposing an AICT algorithm based on continuity to enhance the performance of the RFID file management system. The proposed algorithm is based on an improved collision tree and overcomes the limitation that it can only identify tags with continuous UID distribution.

The algorithm's efficiency and effectiveness are verified through extensive simulations using MATLAB 2017, where it is compared and analyzed with other algorithms such as QT, DBS, CT, and ICT.

The simulation results show that the proposed algorithm outperforms other algorithms regarding identification efficiency and reduces communication traffic during identification. Additionally, it performs well under various tag UID number distribution states and can handle many changes in archive data in unexpected situations. The proposed algorithm successfully realizes the RFID file management system's basic functions, including information collection and processing modules, file management modules, user management modules, and database design.

The application of RFID technology reduces the workload of managers, decreases the risk of file loss and leakage, speeds up the collection efficiency of archives, improves retrieval efficiency, and simplifies the management process. Besides, RFID technology becomes one of the basic technologies needed to realize future smart archives, emphasizing the necessity of research on RFID-based file management systems.

In conclusion, the proposed AICT algorithm based on continuity is a valuable addition to the existing RFID file management systems, and its application could significantly improve the system's performance and efficiency.

Future research will investigate other attributes such as different scales of tag numbers or environmental conditions impacting the efficiency of the proposed algorithm.

### Funding

The authors received no funding for this work.

### Competing Interests

The authors declare there are no competing interests.

### Author Contributions

- Shan Ge conceived and designed the experiments, performed the experiments, analyzed the data, performed the computation work, prepared figures and/or tables, authored or reviewed drafts of the article, and approved the final draft.

### Data Availability

The code is available in the Supplementary File.

### Supplemental Information

Supplemental information for this article can be found online at http://dx.doi.org/10.7717/peerj-cs.1794#supplemental-information.

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
