# Peer review of "Research on intelligent file management system: a design strategy based on RFID technology and improved AICT algorithm"

_PeerJ Computer Science, doi:10.7717/peerj-cs.1794_

## Round 0.1 · original submission · Major Revisions

Dear authors,

Thank you for submitting your article. Reviewers have now commented on your article and suggest major revisions. When submitting the revised version of your article, it will be better to address the following:

1- The research gaps and contributions should be clearly summarized in the introduction section. Please evaluate how your study is different from others in the related section.

2- Please include future research directions.

3- The values for the parameters of the algorithms selected for comparison are not given.

4- The paper lacks the running environment, including software and hardware. The analysis and configurations of experiments should be presented in detail for reproducibility. It is convenient for other researchers to redo your experiments and this makes your work easy acceptance. A table with parameter settings for experimental results and analysis should be included in order to clearly describe them.

5- The authors should clarify the pros and cons of the methods. What are the limitation(s) methodology(ies) adopted in this work? Please indicate practical advantages, and discuss research limitations.

6- Organization of the paper should be given at the end of Introduction section.

7- English grammar and writing style errors should be corrected.

8- Explanation of the equations should be checked. All variables should be written in italic as in the equations.

**Language Note:** The Academic Editor has identified that the English language must be improved. PeerJ can provide language editing services - please contact us at copyediting@peerj.com for pricing (be sure to provide your manuscript number and title). Alternatively, you should make your own arrangements to improve the language quality and provide details in your response letter. – PeerJ Staff

Reviewer 1 ·

Basic reporting

The authors have contributed to the available literature by proposing an algorithm with certain merits. However, the article has issues pertinent to presentation, language, and technical weak points that require more concrete content.

Experimental design

1. All abbreviations should be provided just after the full group of words is presented. Abstract generally does not contain abbreviations.
2. More up-to-date references and discussion are needed in the introduction section.
3. The references in the reference section need to follow a template. All need to be checked.
4. The introduction section should contain two more paragraphs: 1. The research motivation and contribution, 2. The structure of the papers.
5. All titles of sections and subsections should be checked and corrected based on the content where necessary.
6. The whole abstract should be rewritten to summarize what has been researched in the article.
7. Section 2 is very short. Authors should either move it to the introduction section or remove it.
8. There exist some notations in the article such as T4, T8, and Cd. However, no explanations are given. Please check all and fix them.

Validity of the findings

9. The proposed method should be presented in an algorithm.
10. All equations should be cited in the text. Instead of using the word “formula”, the abbreviation, Eq.(.) should be used. The terms in the equations should be explained just below the equations.
11. Which version of MATLAB and its module are used in the simulation?
12. Subsections 4.3.1, 4.3.2, and 4.3.3 should be merged and shortened. The current form has several redundancies.
13. The whole structure of the article should be revised to provide a better presentation.
14. The conclusion section should be rewritten and reorganized since the current form is just a repetition of what has been presented in the previous section.
15. Did the authors run any code and run any statistical analysis to generate the presented results?
16. How were the errors handled? Please discuss it.

Additional comments

please see the above tables for detailed comments.

Reviewer 2 ·

Basic reporting

Authors should use a Professional editing service to enhance the language and presentation of the conducted research. Authors should respond to some technical issues that have put some ambiguity in the research. They need to be clearly explained and answered.

Experimental design

The issues of technicality are itemized as follows:
a. A section should be allocated to the proposed method.
b. All simulations should be better explained and tabulated to be seen better.
c. Did the authors write and run a code based on the steps of the algorithm? Please discuss
d. Which model or modules of MATLAB are used to generate results?
e. What are the simulated data? Please provide more remarks.
f. How does the statistical model work? Please discuss more.
g. Did the authors use any statistical analysis to examine how effectively the proposed method functions? Please discuss more.
h. This sentence is extracted from the text: “Several RFID algorithms were analyzed” What are they? Please provide more remarks and citations.

Validity of the findings

i. What are the metrics used for comparison? Please discuss more and provide more remarks.
j. Did all scenarios cover all possible cases that can be faced in the real system? Please discuss more.
k. Did the authors run pre-processing steps? Please discuss more.
l. What is the error rate of the proposed method? Please provide more information
m. How is the security of transmitted information protected? Please discuss more when personal information is under consideration.

Reviewer 3 ·

Basic reporting

The paper proposing the use of RFID technology to improve file management systems in office environments, highlighting the AICT algorithm's efficiency. However, the English expression in this paper is not yet sufficiently professional, and the references are not well integrated with the text. Although the innovation points and conclusions correspond to each other, the research methods are not well described. Additionally, the field of study is not particularly new, and there is an abundance of related research. It is recommended that the author incorporate more innovative material to enhance the novelty of the paper and consider resubmitting after these improvements are made. There are a few issues that could be addressed to enhance the clarity and effectiveness of the paper:

1. Abstract: The acronym "AICT" is introduced without explaining what it stands for. For clarity, it's important to define acronyms when they are first introduced.
2. Abstract: The compares the AICT algorithm to an "improved collision tree algorithm" but does not provide context about the latter. A brief description of the comparative standard could help readers understand the significance of the improvement.
3. Abstract: The abstract mentions that the algorithm increases recognition efficiency by 10% and reduces communication traffic by 30%. It might be helpful to specify what these percentages are relative to for greater impact.
4. Abstract: The abstract should be reviewed for potential grammatical errors and language refinement to ensure professional and academic rigor.
5. Introduction: While the introduction does a good job of setting the context for the use of RFID technology in file management, it does not directly tie these observations to the specific focus of the paper, which is the design strategy based on RFID technology and the improved AICT algorithm. There is a mention of the drawbacks of traditional file management and the benefits of RFID technology, but the introduction could benefit from a more direct link to how the improved AICT algorithm fits into this narrative. What specific design strategies will the paper propose?
6. Introduction: While the introduction establishes the importance of RFID technology, it should also justify the need for the particular research being conducted. What gap or issue in existing file management systems is the paper addressing?
7. In the 3 section. The text mentions using the Manchester encoding method to identify collision bits but does not provide enough detail on how this encoding is applied within the RFID context and how it contributes to the improvement over other methods.
8. The execution process is outlined in steps, but there could be a clearer flow and linkage between the steps. For example, how does the algorithm determine when to push new query prefixes into the stack pool, and what triggers a switch between binary, quad tree, and octree searches?
9. There is a mention of traffic analysis without specifying what metrics are used to measure traffic or how the improvements affect system scalability and practical deployment.
10. In the 4 section. The phrase "recognition efficiency of 110%" is problematic as efficiency percentages typically should not exceed 100%. This may be a misinterpretation or a misprint that needs to be clarified or corrected.
11. The experiments use a fixed data transmission rate of 96kbps, but it's not clear how this rate was chosen and whether it is representative of typical RFID systems.
12. It's stated that the AICT algorithm performs better in various scenarios, but there is no discussion on the consistency of its performance across different scales of tag numbers or environmental conditions.
13. In the 5 section. The text provides a good level of detail on how the RFID system is intended to function and its various components. However, it does not mention whether this system has been actually implemented and tested in a real-world environment, which would be necessary to validate its effectiveness.
14. While the text mentions avoiding electromagnetic interference by using a multi-channel reader, it does not discuss the potential for RFID signal interference in detail, which is a common problem in densely packed RFID environments.
15. There is a mention of the system prompting management personnel to take timely action if misplaced or missing files are detected, but there are no details on the error handling procedures or how the system supports personnel in resolving such issues.
16. The paper draws conclusions without sufficient discussion of the experimental results. It is recommended to add a section for results and discussion.

The author is advised to consider the aforementioned issues carefully, make the necessary revisions, and then resubmit the paper.

Experimental design

7. In the 3 section. The text mentions using the Manchester encoding method to identify collision bits but does not provide enough detail on how this encoding is applied within the RFID context and how it contributes to the improvement over other methods.
8. The execution process is outlined in steps, but there could be a clearer flow and linkage between the steps. For example, how does the algorithm determine when to push new query prefixes into the stack pool, and what triggers a switch between binary, quad tree, and octree searches?
9. There is a mention of traffic analysis without specifying what metrics are used to measure traffic or how the improvements affect system scalability and practical deployment.
10. In the 4 section. The phrase "recognition efficiency of 110%" is problematic as efficiency percentages typically should not exceed 100%. This may be a misinterpretation or a misprint that needs to be clarified or corrected.
11. The experiments use a fixed data transmission rate of 96kbps, but it's not clear how this rate was chosen and whether it is representative of typical RFID systems.
12. It's stated that the AICT algorithm performs better in various scenarios, but there is no discussion on the consistency of its performance across different scales of tag numbers or environmental conditions.
13. In the 5 section. The text provides a good level of detail on how the RFID system is intended to function and its various components. However, it does not mention whether this system has been actually implemented and tested in a real-world environment, which would be necessary to validate its effectiveness.
14. While the text mentions avoiding electromagnetic interference by using a multi-channel reader, it does not discuss the potential for RFID signal interference in detail, which is a common problem in densely packed RFID environments.
15. There is a mention of the system prompting management personnel to take timely action if misplaced or missing files are detected, but there are no details on the error handling procedures or how the system supports personnel in resolving such issues.

Validity of the findings

The paper draws conclusions without sufficient discussion of the experimental results. It is recommended to add a section for results and discussion.

Additional comments

The author is advised to review more reference literature to substantiate the significance of their research.

Annotated reviews are not available for download in order to protect the identity of reviewers who chose to remain anonymous.

---

## Round 0.2 · accepted · Accept

Dear authors,

Thank you for the revision. It appears that all of the reviewers' comments have been clearly addressed. Your article has been accepted for publication following the final revision.

Best wishes,

Reviewer 1 ·

Basic reporting

The revised paper is well-updated and addresses the comments previously given. therefore, it is recommended for publication

Experimental design

The experiments are well defined and updated so it is recommended for publication

Validity of the findings

The validity is fine and seems to be justified. the comments are incorporated.

Additional comments

The authors have addressed the comments, and there is sufficient for publication.

Reviewer 2 ·

Basic reporting

Thank you for making efforts to address all the raised comments.

Experimental design

no comment

Validity of the findings

no comment

Additional comments

no comment

Reviewer 3 ·

Basic reporting

no comment

Experimental design

no comment

Validity of the findings

no comment